# Local Electric Property Modification of Ferroelectric Tunnel Junctions Induced by Variation of Polarization Charge Screening Conditions under Measurement with Scanning Probe Techniques

**DOI:** 10.3390/nano11123323

**Published:** 2021-12-07

**Authors:** Natalia Andreeva, Anatoliy Petukhov, Oleg Vilkov, Adrian Petraru, Victor Luchinin

**Affiliations:** 1Department of Micro- and Nanoelectronics, St. Petersburg Electrotechnical University ‘LETI’, 197376 Saint Petersburg, Russia; cmid_leti@mail.ru; 2Research Park, St. Petersburg State University, 198504 Saint Petersburg, Russia; anatoliy.petukhov@spbu.ru; 3Department of Solid State Electronics, St. Petersburg State University, 198504 Saint Petersburg, Russia; oleg.vilkov@spbu.ru; 4Nanoelektronik, Technische Fakultät, Christian-Albrechts-Universität zu Kiel, 24143 Kiel, Germany; apt@tf.uni-kiel.de

**Keywords:** ferroelectric tunnel junctions, scanning tunneling spectroscopy in ultrahigh vacuum, tunnel electroresistance

## Abstract

Scanning tunneling spectroscopy in ultrahigh vacuum conditions and conductive atomic-force microscopy in ambient conditions were used to study local electroresistive properties of ferroelectric tunnel junctions SrTiO_3_/La_0.7_Sr_0.3_MnO_3_/BaTiO_3_. Interestingly, experimental current-voltage characteristics appear to strongly depend on the measurement technique applied. It was found that screening conditions of the polarization charges at the interface with a top electrode differ for two scanning probe techniques. As a result, asymmetry of the tunnel barrier height for the opposite ferroelectric polarization orientations may be influenced by the method applied to study the local tunnel electroresistance. Our observations are well described by the theory of electroresistance in ferroelectric tunnel junctions. Based on this, we reveal the main factors that influence the polarization-driven local resistive properties of the device under study. Additionally, we propose an approach to enhance asymmetry of ferroelectric tunnel junctions during measurement. While keeping the high locality of scanning probe techniques, it helps to increase the difference in the value of tunnel electroresistance for the opposite polarization orientations.

## 1. Introduction

Ferroelectric tunnel junctions (FTJs) are considered as non-volatile memory cells with a non-destructive read-out process, ultra-low power consumption, and high storage density. They can also provide a promising solution for an electronic analogue of a synapse for further implementation in neuromorphic systems. In FTJs, two dissimilar electrodes are separated by a thin ferroelectric (FE) film [1,2], the polarization state of which controls the tunnel electroresistance (TER) of the structure. In a classic FTJ, the main transport mechanism is the elastic tunneling. If the top and bottom electrodes are made of different materials, arising asymmetry of the junction changes screening conditions of the polarization charge at the electrode/FE film interfaces. As a result, the potential barrier height varies for the two opposite polarization orientations underlying the switching of the FTJ resistance. Operating characteristics of FTJs are determined by the complexity of observed physical phenomena. Besides the size-effect manifestations, screening of the polarization charge starts to play an important role with decreasing thickness of the FE film. The peculiarities of screening can be tailored by choosing the electrode materials or by changing the conditions of the metal-FE film boundary. Engineering of FTJ-based devices should imply all factors that influence their physical properties, including the impact of interfaces, polarization state, charge-transport mechanisms, and any microscopic inhomogeneity. Therefore, the point issue arises on relevant techniques to study the local electrophysical and structural properties of heterosystems with FTJs in order to verify optimal operational parameters for target-device integration into neuromorphic circuits.

One of the most convenient techniques to investigate the correlation between local structural, resistive, ferroelectric properties of thin FE films is atomic force microscopy (AFM). It is considered that combining the regimes of tunnel AFM, used for measuring small currents through FE film, and piezoresponse force microscopy to take local piezoresponse from FE films, one could reveal the ferroelectric/resistive property distribution over the film surface topography with spatial resolution down to the nanometer scale.

A simple estimation of the AFM probe/FE film contact sizes suggests that even at relatively large curvature radius of AFM probes (several tens of nanometers, which is typical for conductive AFM probes), the measured current density for low (*V* ≈ 0) and intermediate (*V* ≤ *φ*/*e*, where *φ* is the barrier height at the interface, *e*—charge of electron) voltages applied to the tunnel junctions should be well below the sensitivity (noise) level of the standard AFM setup. Moreover, the influence of the water layer adsorbed on the FE film surface significantly distorts the screening conditions of polarization charges [3,4,5], which could bring us to a misleading interpretation of the data. The situation does not get much better by switching to high-vacuum AFM measurements, as the contact radius becomes smaller, making the effects of electrostatic interactions between the AFM cantilever and the sample surface more apparent. Thus, the only method for the reliable measurement of TER is to place the AFM probe on the deposited top electrode of the FTJ, reducing spatial resolution and considerably complicating the measurement of correlated FE properties.

Recently, we proposed a new approach based on scanning tunneling microscopy/spectroscopy under ultrahigh vacuum (UHV) conditions to study the influence of FE film structure on the local resistive properties of thin FE films, including resistive switching effects [6]. In this work, we extend this approach to investigate the correlation between resistive and ferroelectric properties in FTJs. The angstrom-sized tunnel barrier between the STM probe and the FE film surface induces a strong asymmetry in polarization charge screening at the “FE film/STM probe” and “FE film/bottom electrode” interfaces. In this case, the TER ratio for opposite polarization states increases compared to that of the FTJ devices with deposited top electrodes. UHV conditions minimize the influence of surface adsorbate on both the physical properties of FTJs and the results of measurements, while the current sensitivity and spatial locality of STM measurements are significantly higher than those of AFM.

The tunnel current feedback organization in STM ensures the stability of “STM probe/surface” contact for electric measurements, compared with that of AFM. The polarization state in FE thin film is switched by voltage pulses applied between the bottom electrode of the FTJ and STM probe, while the *I-V* curves measurements are used for evaluation of local resistive properties.

## 2. Materials and Methods

La_0.7_Sr_0.3_MnO_3_/BaTiO_3_ (LSMO/BTO) heterostructures were deposited on (001)-oriented single-crystalline SrTiO_3_ (STO) substrates. Commercially available STO STEP substrates (Shinkosha), possessing an excellent polished surface with flat atomic/molecular step terraces, were used in this study. LSMO (10 nm)/BTO (2.5 nm) heterostructures were grown via PLD on the mentioned substrates. The deposition process was monitored in situ with reflection high-energy electron diffraction (RHEED) (see the Supplementary File in Ref. [6]). RHEED intensity oscillations indicate a layer-by-layer growth of LSMO. During the growth of BTO films, no RHEED oscillations were observed, but the RHEED pattern remained that of a two-dimensional surface, indicating changes of the growth mode to a step flow. After LSMO/BTO deposition, the samples were cooled down to room temperature under higher oxygen pressure. Gold top electrodes with thicknesses of 10–15 nm were deposited on the BTO films by thermal evaporation. The top electrodes were patterned using a stencil mask, the area of which was in the 2 μm^2^ to 36 μm^2^ range. X-ray diffraction (XRD) experiments were carried out with a SmartLab difractometer (Rigaku) equipped with a 9 kW Cu anode X-ray tube. The results of the in-plane XRD scan confirmed that the LSMO and the BTO films were fully strained on the STO substrate, since the in-plane lattice of the BTO film equals that of the STO substrate (see the Supplementary File in Ref. [6]).

The ferroelectric properties of fabricated FTJs were studied at room temperature in ambient conditions, employing piezoresponse force microscopy (PFM) technique of a Veeco SPM instrument. The hysteresis loops and the results of the polarization procedure obtained with PFM are given in Figure 1. The FE hysteresis loops are symmetric and rectangular; the coercive bias is 2 V according to the hysteresis loops.

*I-V* curves of FTJs were measured through the top gold electrode with conductive AFM tip by applying quasi-static voltage sweeps. Solid platinum AFM probes (25Pt300A, Rocky Mountain Nanotechnology LLC, Holladay, UT, USA) were used. Voltage was applied to the bottom electrode of the structure with the probe grounded. To eliminate a possible influence of hysteretic effects (originated from the transient processes in ferroelectric) on the shape of experimental *I-V* curves for FTJs with different polarization states, the scan rate of *I-V* measurements was set to 0.01 Hz, based on the estimations of the FTJ capacitance:(1)$$C=\frac{\varepsilon \cdot \varepsilon_{0}\cdot S_{TE}}{d}=0.01\;÷0.1\;nF,$$
where $\varepsilon_{0}$ is permittivity of free space, ε is the dielectric constant of the FE film, *S_TE_* is area of the top electrode and d is film thickness. For 2.5 nm–thick FE film with 36 μm^2^ top electrode areas and ε = 100–1000, the resistance of FTJ, measured with a low (0.1 V) dc voltage, is 500 MΩ. The time constant for FTJ (5·τ=5·R·C) is within the range of 25 ÷250 ms, limiting the scan rate of *I-V* measurements by the value of sampling time in the range of 200–20 ms per point in case of AFM. This corresponds to a scan rate of *I-V*-measurements of 10 mV s^−1^−25 mV s^−1^.

FE polarization switching was induced by voltage pulses of 3.0 V amplitude and 1 s duration, which were applied to the bottom electrode of FTJs, while an AFM probe, placed on the top electrode, was grounded. Changing the polarity of the applied voltage gives rise to the corresponding polarization reorientation in the film.

The STM/STS measurements were performed under ultrahigh vacuum (UHV) conditions at the resource center “Physical Methods of Surface Investigation” (RC PMSI) of Research Park of Saint Petersburg State University. The base pressure in the UHV chamber was better than 3 × 10^−10^ mbar. Residual gas particles adsorbed on a sample surface were removed by heating up to 120 °C in UHV prior to measurement.

*I-V* curves were obtained in the tunneling current range from 1 pA to 330 nA using an Omicron VT AFM XA scanning probe microscope. The scan rate and sampling times for *I-V* measurements correspond to the previously estimated values, in accordance with FTJ device capacitance. For STM/STS measurements, tungsten STM tips were used. FE polarization switching was induced by voltage pulses of 5 V amplitude and 1 s duration, which were applied between the STM tip and the LSMO bottom electrode.

## 3. Results and Discussions

For measurements of TER ratio, induced by polarization switching in thin BTO films, we first consider the FTJs with a gold top electrode and AFM probe placed on it (Figure 2a). Polarization charges at the interfaces with electrodes are compensated by screening charges in electrodes distributed over the electronic screening length. This screening length could be estimated, for example, in accordance with Thomas-Fermi model [7,8], where the electric field is attenuated over a distance, which is inversely proportional to the Thomas-Fermi wave vector: $k_{TF}={\left[4\pi e^{2}g\left(\varepsilon_{F}\right)\right]}^{1/2}$,$\;g\left(\varepsilon_{F}\right)$ is the density of states at the Fermi level. Estimations based on the band structure calculations show that in the case of the FE film/metallic electrode interface, the screening length could be as much as 3 Å for PZT/LSMO [8]. Considering our samples, it is more realistic if FE film in FTJs possesses complex interfacial bonds, and the potential drop near the interface should be described by the effective screening length and dielectric response [9] in terms of λeffε, where $\lambda_{eff}$ is an effective screening length for the FE film/electrode interface and ε is relative dielectric constant. For the LSMO/BTO interface, effective screening length and relative dielectric constant could be accepted as equal to 1 Å [9,10] and 9.6 [9], correspondingly. Thus, for the LSMO/BTO interface, λeffBE=λeffε could be estimated as 0.1 Å. For BTO/Au interfaces, substantially better electronic screening was predicted to result in dead-layer-free capacitors [11]. In realistic first-principles calculations, the inverse permittivity profile for BTO/Au interface is comparable to that for a BTO/Pt interface [12], and for the initial guess in our model, we took the value of $\lambda_{eff}^{TE}=0.003\;Å$, obtained for a BTO/Pt interface in the framework of the “series capacitor model”, in which the dielectric constant, ε, has already been taken into account [12].

On the basis of continuity of electric displacement, under short-circuit boundary conditions, the finite depolarization field, ${Ε}_{d}$, in the film associated with the imperfect screening of the polarization charges [13] is given by:(2)$${Ε}_{d}=-\frac{1}{d}\frac{P_{s}}{\varepsilon_{0}}\left(\lambda_{eff}^{BE}+\lambda_{eff}^{TE}\right),$$
where $P_{s}=$26 μC/cm^2^ [10] is the value of spontaneous polarization for thin BTO film, *d* = 25 Å is the film thickness, and $\lambda_{eff}^{TE}=0.003\;Å$ and $\lambda_{eff}^{BE}$ = 0.1 Å are effective screening lengths for the BTO/Au top electrode and LSMO bottom electrode/BTO interfaces, respectively. The depolarization field in this case is $E_{d}=4.6\cdot 10^{5}$ V/cm^2^, and the voltage drops at the interface with bottom and top electrodes are $∆V^{BE}=0.293\;V$ and $∆V^{TE}=0.001\;V$, correspondingly.

At small-reading voltage, bias-dependent tunnel resistance could be described by the Brinkman model [14]. However, the Brinkman model is applicable for the parabolic dependence of conductance on voltage for the voltages below ≤0.5 V and within the offset of parabolic dependence below 250 mV. In this work, taking into account the voltages practically applied to FTJs in AFM conductive measurements, we refer to the theory of Simmons for the electric tunnel effect between dissimilar electrodes separated by a thin insulating film [15]. According to the Simmons approach, for a trapezoidal barrier in the intermediate voltage range 0<V≤ φe, the current-voltage characteristics are independent of bias polarity, and the density of current from electrode 1 to electrode 2 is given by:(3)$$J_{1}=\left(\frac{e}{4{\pi \mathrm{hd}}^{2}}\right)\{\left(\varphi_{1}+\varphi_{2}-\mathrm{eV}\right)\times \exp\left[-\left(\frac{4{\pi \mathrm{dm}}^{1/2}}{h}\right){\left(\varphi_{1}+\varphi_{2}-\mathrm{eV}\right)}^{1/2}\right]\;\;\;\;\;\;\;\;\;\;\;-\left(\varphi_{1}+\varphi_{2}+\mathrm{eV}\right)\times \exp\left[-\left(\frac{4{\pi \mathrm{dm}}^{1/2}}{h}\right){\left(\varphi_{1}+\varphi_{2}+\mathrm{eV}\right)}^{1/2}\right]\},$$
where *e* and m are charge and mass of electron, *h* is Planck’s constant, *V* is voltage across the film, and *φ*_1(2)_ is barrier height at the interface of electrode 1 (2) and FE film.

In the case of thin FE film, at the interface with top and bottom electrodes, the drop of the voltage due to the depolarization field is given by:(4)$$\varphi_{1}^{↓}=\varphi_{1}-∆V^{BE}\;\mathrm{and}\;\varphi_{2}^{↓}=\varphi_{2}+∆V^{TE},$$
for the downward orientation of the FE polarization, i.e., in the direction to the bottom (LSMO) electrode (Figure 2c,d), and for the upward orientation of the polarization:(5)$$\varphi_{1}^{↑}=\varphi_{1}+∆V^{BE}\;\mathrm{and}\;\varphi_{2}^{↑}=\varphi_{2}-∆V^{TE}.$$

This results in the corresponding current density:(6)$$J_{1}^{↓\left(↑\right)}=\left(\frac{e}{4\pi hd^{2}}\right)\left\{\left(\varphi_{1}^{↓\left(↑\right)}+\varphi_{2}^{↓\left(↑\right)}-eV\right)\times exp\left[-\left(\frac{4\pi dm^{1/2}}{h}\right){\left(\varphi_{1}^{↓\left(↑\right)}+\varphi_{2}^{↓\left(↑\right)}-eV\right)}^{1/2}\right]\;\;\;\;\;\;\;\;\;\;-\left(\varphi_{1}^{↓\left(↑\right)}+\varphi_{2}^{↓\left(↑\right)}+eV\right)\times exp\left[-\left(\frac{4\pi dm^{1/2}}{h}\right){\left(\varphi_{1}^{↓\left(↑\right)}+\varphi_{2}^{↓\left(↑\right)}+eV\right)}^{1/2}\right]\right\}$$

Let us consider the high-voltage range, V>φe, and asymmetrical barrier, as the gold electrode has a lower work function: ψ_Au_ = 4.76 eV, ψ_LSMO_ = 4.8 eV. If an electrode with a lower work function is negatively biased (reverse polarity of applied bias), the voltage range starts from V >φ1e, and density of the current, flowing from this electrode, is described by:(7)$$J_{1}=\frac{1.1e{\left(eV-∆\varphi \right)}^{2}}{4\pi h\varphi_{2}d^{2}}\left\{exp\left[\left(-\frac{23\pi m^{1/2}}{6h}\right)\left(\frac{d\varphi_{2}^{3/2}}{eV-∆\varphi }\right)\right]\;\;\;\;\;\;\;\;\;\;\;\;\;\;\;\;\;-\left(1+\frac{2eV}{\varphi_{2}}\right)\times exp\left[\left(-\frac{23\pi m^{1/2}}{6h}\right)\left(\frac{d\varphi_{2}^{3/2}{\left[1+\left(2eV/\varphi_{2}\right)\right]}^{1/2}}{eV-∆\varphi }\right)\right]\right\}$$

For the forward polarity of applied bias, when the electrode with a lower work function is positively biased, the voltage range starting from V >φ2e and the current density are given by:(8)$$J_{2}=\frac{1.1e{\left(eV+∆\varphi \right)}^{2}}{4\pi h\varphi_{1}d^{2}}\left\{exp\left[\left(-\frac{23\pi m^{1/2}}{6h}\right)\left(\frac{d\varphi_{1}^{3/2}}{eV+∆\varphi }\right)\right]\;\;\;\;\;\;\;\;\;\;\;\;\;\;\;\;\;-\left(1+\frac{2eV}{\varphi_{1}}\right)\times exp\left[\left(-\frac{23\pi m^{1/2}}{6h}\right)\left(\frac{d\varphi_{1}^{3/2}{\left[1+\left(2eV/\varphi_{1}\right)\right]}^{1/2}}{eV+∆\varphi }\right)\right]\right\}$$

With the FE polarization taken into account, for the downward (upward) polarization and at high voltages, V $>\frac{\min\left(\varphi_{2}+∆V^{TE},\;\varphi_{1}-∆V^{BE}\right)}{e}$ (V $>\frac{\min\left(\varphi_{2}-∆V^{TE},\;\varphi_{1}+∆V^{BE}\right)}{e}$), the reverse characteristics (at the reverse bias) are described by:(9)$$\begin{array}{c} J_{1}^{↓\left(↑\right)}=\frac{1.1e{\left(eV-∆\varphi^{↓\left(↑\right)}\right)}^{2}}{4\pi h\varphi_{2}^{↓\left(↑\right)}d^{2}}\{exp\left[\left(-\frac{23\pi m^{1/2}}{6h}\right)\left(\frac{d{\left(\varphi_{2}^{↓\left(↑\right)}\right)}^{3/2}}{eV-∆\varphi^{↓\left(↑\right)}}\right)\right]\\ \;\;\;\;\;\;\;\;\;\;\;\;\;-\left(1+\frac{2eV}{\varphi_{2}^{↓\left(↑\right)}}\right)\\ \;\;\;\;\;\;\;\;\;\;\;\;\;\times exp\left[\left(-\frac{23\pi m^{1/2}}{6h}\right)\left(\frac{d{\left(\varphi_{2}^{↓\left(↑\right)}\right)}^{3/2}{\left[1+\left(2eV/\varphi_{2}^{↓\left(↑\right)}\right)\right]}^{1/2}}{eV-∆\varphi^{↓\left(↑\right)}}\right)\right]\}, \end{array}$$
where $∆\varphi^{↓}=\varphi_{1}^{↓}-\varphi_{2}^{↓}=(\varphi_{1}-∆V^{BE})-(\varphi_{2}+∆V^{TE})$ is for downward polarization and $∆\varphi^{↑}=\varphi_{1}^{↑}-\varphi_{2}^{↑}=\left(\varphi_{1}+∆V^{BE}\right)-(\varphi_{2}-∆V^{TE})$ is for upward polarization.

For the downward (upward) polarization and voltage range, V $>\frac{\min\left(\varphi_{2}+∆V^{TE},\;\varphi_{1}-∆V^{BE}\right)}{e}$ (V $>\frac{\min\left(\varphi_{2}-∆V^{TE},\;\varphi_{1}+∆V^{BE}\right)}{e}$), the forward characteristics (at the forward bias) are given by:(10)$$\begin{array}{c} J_{2}^{↓\left(↑\right)}=\frac{1.1e{\left(eV+∆\varphi^{↓\left(↑\right)}\right)}^{2}}{4\pi h\varphi_{1}^{↓\left(↑\right)}d^{2}}\{exp\left[\left(-\frac{23\pi m^{1/2}}{6h}\right)\left(\frac{d{\left(\varphi_{1}^{↓\left(↑\right)}\right)}^{3/2}}{eV+∆\varphi^{↓\left(↑\right)}}\right)\right]\\ \;\;\;\;\;\;\;\;\;\;\;\;-\left(1+\frac{2eV}{\varphi_{1}^{↓\left(↑\right)}}\right)\\ \;\;\;\;\;\;\;\;\;\;\;\;\times exp\left[\left(-\frac{23\pi m^{1/2}}{6h}\right)\left(\frac{d{\left(\varphi_{1}^{↓\left(↑\right)}\right)}^{3/2}{\left[1+\left(2eV/\varphi_{1}^{↓\left(↑\right)}\right)\right]}^{1/2}}{eV+∆\varphi^{↓\left(↑\right)}}\right)\right]\},\; \end{array}$$

The modeled *I-V* characteristics of FTJs with 25 Å–thick BTO film with top gold and bottom LSMO electrodes are given in Figure 2b. A description of the parameters used for *I-V* curve modeling is shown in Table 1. For the intermediate voltage range, the *I-V* characteristics are independent of the bias polarity, and at given values of effective screening lengths for the top and bottom electrodes, switching the FE polarization in thin film results in one or two orders of difference in the values of TER for downward and upward polarization. Relatively small difference for the reverse and forward *I-V* characteristics is expected due to the small difference in electrode work functions. It should be mentioned that the values of the current for the modeled *I-V* curves are obtained for 28 μm^2^-sized top electrodes. In this case, the current level at relatively small voltages (below 0.5 V) does not exceed 10 nA. In conventional measurements of current maps from FTJs with conductive AFM (see, for ex. [16]), when the tunnel current is measured with the AFM tip serving as a top electrode, the high-voltage range is used. The size of “FE film–AFM tip” contact area is restricted by the curvature radius of the AFM tip and does not exceed a few tens of squared nanometers. This considerably decreases the current below the noise level of the conductive AFM technique in low and intermediate voltage ranges.

The results of *IV*-measurements with an AFM tip placed on a top gold electrode of an STO/LSMO/BTO sample are presented in Figure 3a. The voltage range applied to the structure was chosen in accordance with experimental FE-hysteresis loop and was restricted by 1 V to not exceed the coercive voltage of the FE film. Experimental data taken at the downward polarization after applying a negative voltage pulse to the bottom electrode are depicted with red asterisks, while blue cross-like markers are for the upward polarization. To explain the relatively small changes in the resistance after polarization reorientation, experimental data were approximated by the curve, simulating the tunnel current through the structure with dissimilar electrodes and imperfect polarization charge screening. For the given FE film thickness and electrode materials, we fixed the potential barrier at the LSMO bottom electrode/BTO interface, as well as the corresponding drop of voltage, because both the LSMO electrode and BTO thin film were deposited consistently in one PLD cycle, and they were fully strained on the STO substrate, which was confirmed by XRD. Effective screening length and the potential barrier height at the top gold electrode were chosen as fitting parameters. This seems reasonable, as the residual water layers could be present on the surface of the oxide film during gold-electrode deposition [14,17]. Physically and chemically adsorbed water layers could not be completely removed from the oxide surface, even after annealing at temperatures as high as 350 °C, and they were able to significantly influence the screening conditions of the polarization charges at the interface with the top electrode. The results of our approximation are depicted by dashed red curves for downward polarization and by dotted curves for upward polarization. A good correlation with the experiment was observed for the voltage drop of 0.195 V, associated with imperfect polarization screening at the interface with the top electrode. Corresponding effective screening length was 0.07 Å (instead of 0.003 Å). Moreover, the significant difference in experimental *I-V* curves for reverse and forward polarities indicates that the barrier height asymmetry at the interfaces with top and bottom electrodes was initially underestimated. According to the results of the approximation, the barrier height at the interface with top electrode is sufficiently higher than was assumed based on the work function difference. $\varphi_{Au}=\varphi_{LSMO}+\left(\Psi_{Au}-\Psi_{LSMO}\right)=0.86\;\mathrm{eV}$. In practice, it equals 1.4 eV, which could be explained by the presence of an additional adsorbate layer at the interface with the top electrode.

Interestingly, the results of *I-V* curve modeling for the high-voltage range (given in Figure 3b) indicate the appearance of *I-V* curve twisting for opposite polarization orientations under transition from the intermediate to high-voltage range. At intermediate voltages, the *I-V* curves for upward polarization lie above (below) the curves for the downward polarized state for the reverse (forward) polarity of applied bias. At a high-voltage range, the situation is opposite. The reason for this curve twisting is a combination of (i) high barrier height asymmetry at the interfaces and (ii) small difference in effective screening lengths of polarization charges at the electrodes. In Figure 3c, experimental evidence of such a twisting is demonstrated.

In STM measurements, the probe serves as a top electrode (STM-TE). The tunnel barrier existing between the probe and FE film surface (Figure 4a) increases the height of the potential barrier at the “FE film–top electrode” interface and influences the polarization charge screening. As we suppose, such a screening occurs inside the tunnel barrier between the probe and the FE film surface and is related to OH groups chemically bonded to the BTO surface [5]. In contrast with physisorbed OH or H_2_O layers, which can be removed by annealing under UHV conditions, chemically adsorbed layers remain at the surface, even at 800 °C [17]. Thus, for two opposite polarization orientations, the voltage drop associated with polarization charge screening in UHV does not give a reasonable impact to the value of an existing potential barrier at the interface with STM-TE (Figure 4c,d). The height of this barrier could be retrieved from experimental *I-V* characteristics based on the energy diagram of the barrier (Figure 4c). In this way, for the reverse direction of a junction with a negatively biased LSMO electrode, the *I-V* characteristics for downward and upward polarizations cross over at the voltage near the potential barrier height at the interface with STM-TE.

Theoretical *I-V* curves calculated for STO/LSMO/BTO structure with a 25 Å-thick FE film and tungsten STM tip and the potential barrier, $\varphi_{Tun.Bar.}=2.5\;V$, at the interface with STM-TE are given in Figure 4b. The current density was obtained for the curvature radius of the tip, R ≈ 16 nm. It should be noted that the lateral resolution of the STM in a scanning regime is typically much higher than 16 nm, and it can be estimated as ${\left[\left(R+s\right)\left[Å\right]\right]}^{1/2}$ [18,19], where s is a distance between the tip and the surface. The value of R = 75 Å was obtained from scanning electron microscope (SEM) images of the STM tip taken in backscattered-electron-detection mode (AsB) with a beam energy of 20 kV and a beam current of 281 pA. For typical distances, s = 10 Å between the STM tip and surfaces of dielectric thin films. The upper limit of the resolution can be estimated as 1 nm at R + s = 85 Å. Nevertheless, at spectroscopic measurements, the larger tip radii ensure better stability of a tunnel current. This is especially important at room temperature, providing the value of the current above the noise level of the setup.

Compared with AFM results, *I-V* characteristics obtained with STM exhibit increasing asymmetry for the forward and reverse directions of applied bias. This arises from the larger value of ∆φ = $\varphi_{Tun.Bar.}-\varphi_{LSMO}$ (Figure 4b). More importantly, the greater difference in the values of the current through the junction for upward and downward polarizations is observed in STM. This fact can be considered an important benefit of the STM spectroscopic technique over the conductive AFM technique in studying the local conductive properties of thin FE films.

The resistance of the STM tunnel barrier should be considered a series resistance when experimental data are analyzed. The value of this resistance can be estimated from the linear part of experimental *I-V* curves. In our case, this approach yields  Rtun=1V10 pA=1011Ω. Then, the relationship between the current density and the voltage is given by: $J_{1,2}^{↓\left(↑\right)}=f\;\left(V-J_{1,2}^{↓\left(↑\right)}\times R_{tun}\right)$. In practice, the tunnel barrier acts as a limiting resistor, reducing the current in measurements. The effect of this resistance on the current-voltage characteristics of STO/LSMO/BTO FTJs is shown in Figure 5a. There, the calculated *I-V* curves for upward polarization are depicted by dotted (without contact resistance) and short-dotted (with contact resistance) lines, and for downward polarization, by dashed and short-dashed lines, correspondingly. It should be noted that the voltage drop at the tunnel contact resistance forced us to increase the amplitude of voltage pulses applied to FTJs for polarization switching up to 5 V.

The results of *IV*-measurements of STO/LSMO/BTO structures with an STM tip are given in Figure 5b. The voltage range applied to the structure was chosen in accordance with the resistance of the “STM tip-FE film” tunnel barrier and did not exceed the coercive voltage of FE film. It was restricted by 2 V for the forward *I-V* curve and by 3 V for the reverse characteristics. The curves taken for downward polarization after applying the positive voltage pulse to the STM tip are depicted with red asterisk markers, while the plus-shaped blue markers are for upward polarization. The approximation of experimental *I-V* curves for 25 Å-thick FE film with effective screening length $\lambda_{eff}^{BE}$ = 0.1 Å and voltage drop $∆V^{BE}=0.293\;V$ at the interface with LSMO bottom electrode are presented in Figure 5b. The potential barrier height at the top electrode and the tunnel contact resistance were chosen as fitting parameters. The best agreement with the experiment was observed for the values of $\varphi_{Tun.Bar.}=2.1\;V$ and $R_{tun}=2.6\times 10^{11}\Omega$. This correlates well with initially accepted assumptions and confirms that the chosen approach can be successfully applied for TER investigation in thin FE films with the STM technique.

The difference between *I-V* curves for upward and downward polarizations is generally larger in the case of STM measurements compared to AFM measurement. This is due to the increased difference in the combined “FE film + STM” tunnel barrier height for opposite polarization orientations. The reason for the higher asymmetry in barrier heights is the specificity of polarization charge screening occurring inside the STM tunnel barrier. This minimizes the impact of the imperfections at the “FE film–deposited top electrode” interface on the polarization charge screening taking place in AFM measurements. In the case of AFM, the non-ideal “FE film–top electrode” interface increases the effective screening length and, in practice, reduces the difference in TER for opposite polarization orientations.

## 4. Conclusions

In summary, local resistive properties of STO/LSMO/BTO structures were investigated with AFM and STM techniques to reveal the peculiarities of polarization charge screening and its effect on the value of TER. The difference in *I-V* curves of FTJs, taken for opposite polarizations with an AFM tip placed on top of a gold electrode, appears to be much less than that expected from theoretical considerations based on the evaluation of the mean barrier height for upward and downward polarization states. The observed discrepancy between experimental and theoretical *I-V* curves uncovers enhanced effective screening length at the interface of FE film with the top Au electrode. This can be explained by the presence of OH groups chemically bonded to the BTO surface. In the case of STM measurements with a tip placed on the FE film surface, asymmetry in the mean barrier height for different polarization states is more pronounced compared with AFM measurements since polarization charge screening occurs inside the STM tunnel barrier, raising the TER ratio of the junctions. Besides, STM offers better locality of measurements and higher current sensitivity than AFM. The presence of an additional tunnel barrier in STM measurements does not shift the device to the Fowler-Nordheim tunneling regime, and the applied voltages for IV measurements remain below the coercive voltage for polarization switching in FE film.

## Figures and Tables

**Figure 1 nanomaterials-11-03323-f001:**
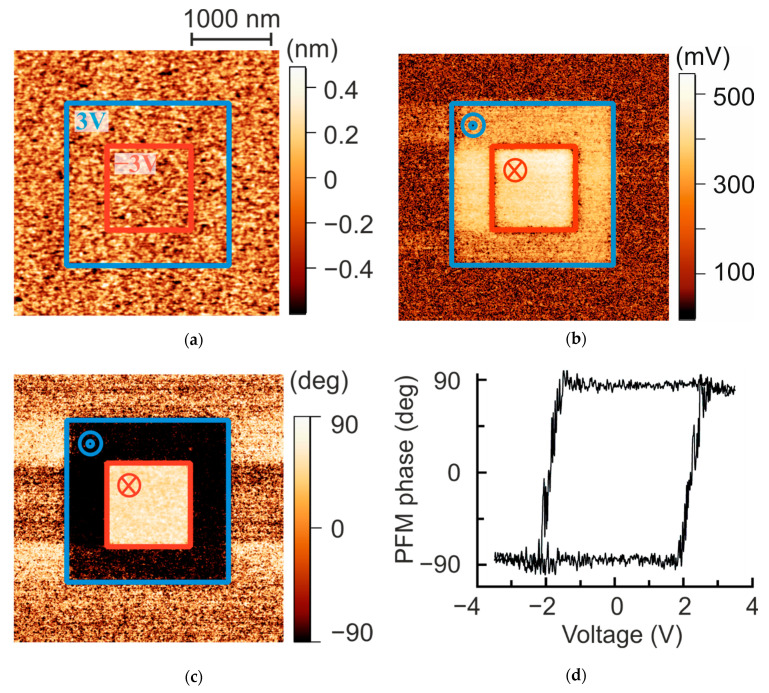
FE properties of STO/LSMO/BTO heterostructures measured with PFM at room temperature in ambient conditions. (**a**) Topography (3.5 × 3.5 μm^2^); (**b**) PFM out-of-plane amplitude and (**c**) phase image after poling procedure with DC bias applied to the LSMO bottom electrode, the square domains with downward and upward polarizations written with ±3 V. PFM imaging was performed with 399 kHz AC voltage of 400 mV applied to the bottom electrode; (**d**) local piezoresponse-voltage hysteresis loop measured on bare BTO film surface in spectroscopic PFM regime.

**Figure 2 nanomaterials-11-03323-f002:**
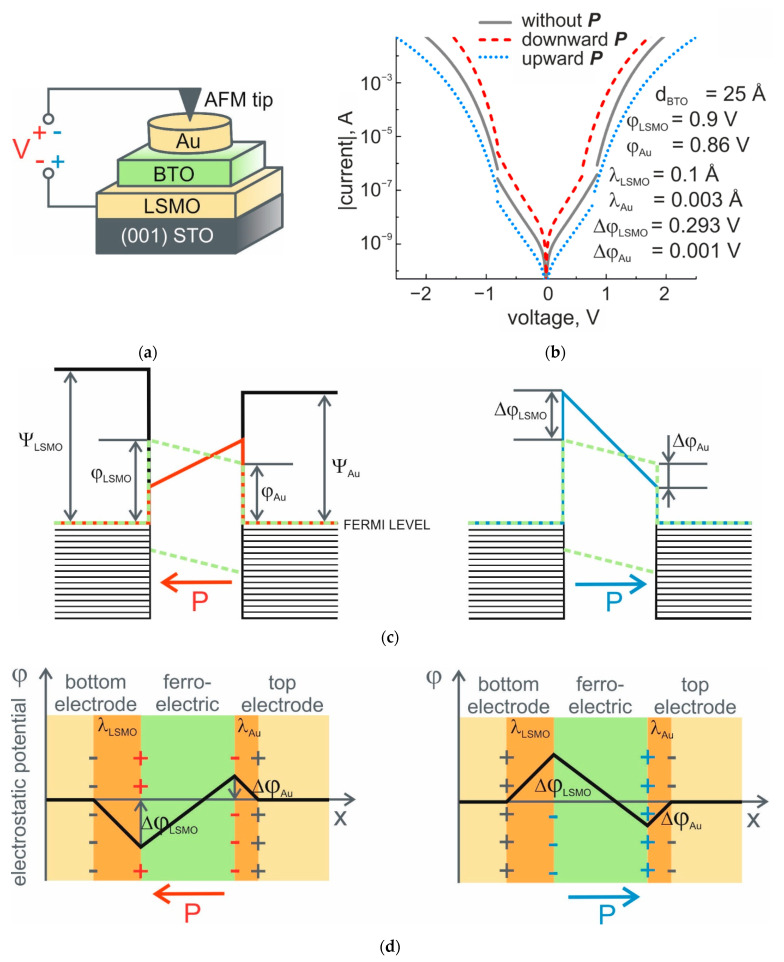
Conductive AFM studies of TER. (**a**) Schematic illustration of a STO/LSMO/BTO/Au FTJ and a method for investigation of resistive properties with AFM tip placed on top of gold electrodes; (**b**) simulated *I*-*V* curves of FTJ with dissimilar electrodes and polarization charge compensation taken into account. The simulation is based on the electrostatic model and the Simmons approach for intermediate and high-voltage ranges; (**c**) energy diagram of the barrier between dissimilar electrodes at different polarization orientations; (**d**) schematic representation of the residual uncompensated potential, *φ(x)*, for uniformly polarized FE film between two dissimilar electrodes with effective screening lengths *λ_LSMO_* and *λ_Au_* for two polarization states. Left to right: for polarization pointing to the left and for polarization pointing to the right.

**Figure 3 nanomaterials-11-03323-f003:**
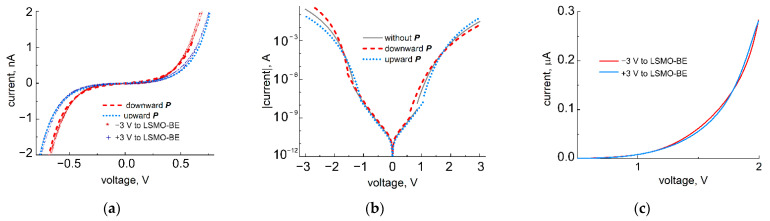
Experimental (depicted with red asterisks and blue cross-like markers for downward and upward polarization, correspondingly) and theoretical (depicted with red dashed and blue dotted curves for downward and upward polarization, correspondingly) *I-V* characteristics of the STO/LSMO/BTO structures with 25 Å–thick FE film for two opposite polarization orientations. (**a**) *I-V* curves measured with AFM tip placed on a top gold electrode at intermediate voltages and fitted in accordance with the Simmons approach and polarization charge compensation with $∆V^{BE}=0.293\;V,\;∆V^{TE}=0.195\;V,\;\varphi_{LSMO}=0.9\;V,\;\varphi_{Au}=1.4\;V$; (**b**) theoretical *I-V* curves extended for the high-voltage range; the positive polarity of applied bias corresponds to the current flowing in the forward direction, while the negative polarity applies to the reverse direction; (**c**) experimental *I-V* curve twisting for opposite polarization orientations under transition from intermediate to high-voltage range, measured with AFM tip placed on Au top electrode of the STO/LSMO/BTO structure.

**Figure 4 nanomaterials-11-03323-f004:**
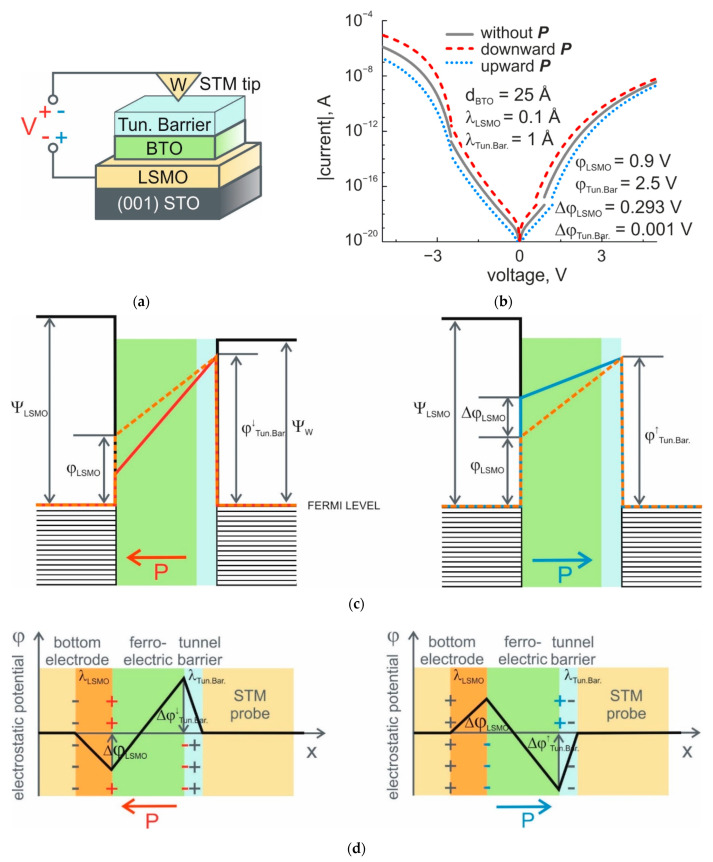
STM study of TER. (**a**) Schematic illustration of a STO/LSMO/BTO FTJ and a method for investigation of the resistive properties with STM tip; (**b**) simulated I-V curves of FTJs in case of STM measurements. The simulation is based on the imperfect screening of polarization charges at the interface with bottom electrode and the Simmons approach for intermediate and high-voltage ranges; (**c**) energy diagram of the barrier consisting of thin FE film and STM tunnel gap between the bottom electrode of the structure and the STM tip at different polarization orientations; (**d**) schematic representation of the residual uncompensated potential, φ(x), for uniformly polarized FE film between bottom electrode and STM tip with effective screening length, λ_LSMO_, and width of a tunnel gap, λ_Tun.Bar._, for two polarization states. Left to right: for polarization pointing to the left and for polarization pointing to the right.

**Figure 5 nanomaterials-11-03323-f005:**
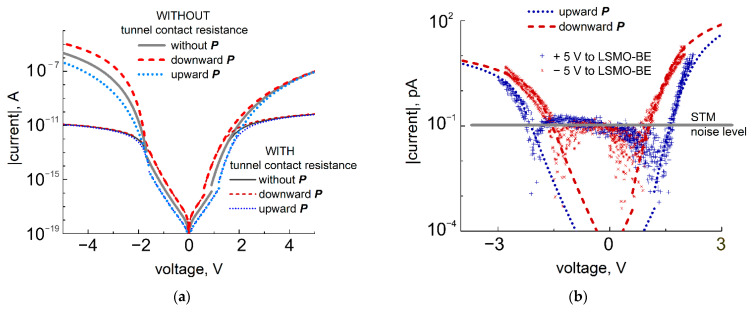
Experimental and theoretical *I-V* characteristics of the STO/LSMO/BTO structures with 25 Å–thick FE film for two opposite polarization orientations in case of STM studies. (**a**) Theoretical *I-V* curves extended for the high-voltage range; the positive polarity of applied bias corresponds to the current flowing in the forward direction, while the negative polarity corresponds to the reverse direction. (**b**) *I-V* curves measured with STM tip placed on the surface of BTO film at intermediate voltages and fitted in accordance with the Simmons approach and polarization charge compensation with $∆V^{BE}=0.293\;V,\varphi_{LSMO}=0.9\;V,\;\;\varphi_{Tun.Bar}=2.1\;V,\;R_{tun}=2.6\times 10^{11}\;\Omega$.

**Table 1 nanomaterials-11-03323-t001:** Parameters for modelling I-V characteristics in measurements with AFM and STM techniques.

Parameter	Description	Default Value
*d*	BTO film thickness	25 Å
*S_Au_*	Areas of the Au top electrode	28 μm^2^
*S_STM_*	Areas of the STM contact	10 nm^2^
ψ_LSMO_	Work function of LSMO bottom electrode	4.8 eV
ψ_Au_	Work function of Au top electrode	4.76 eV
*φ_1_ = φ_LSMO_*	Potential barrier height at the interface with LSMO bottom electrode	0.9 V
*φ_2_ = φ_Au_*	Potential barrier height at the interface with Au top electrode in AFM measurements	0.86 V
*φ_Tun.Bar._*	Potential barrier height at the interface with STM probe serving as a top electrode in STM measurements	2.5 V
*Δφ_LSMO_*	Voltage drop at the LSMO bottom electrode/BTO interface associated with imperfect polarization	0.293 V
*Δφ_Au_*	Voltage drop at the Au top electrode/BTO associated with imperfect polarization in AFM measurements	0.001 V
*λ_LSMO_*	Effective screening length for the LSMO bottom electrode/BTO interface	0.1 Å
*λ_Au_*	Effective screening length for the Au top electrode/BTO interface	0.003 Å
*P_s_*	Spontaneous polarization	0.26 C m^−2^
*R_tun_*	Resistance of STM tunnel barrier	10^11^ Ω
Constant	Description	Value
*m*	Electron mass	9.11 × 10^−31^ kg
*e*	Elementary charge	1.6 × 10^−19^ C
*h*	Planck’s constant	6.63 × 10^−34^ J s
ε_0_	Vacuum dielectric constant	8.85 × 10^−12^ F m^−1^

## Data Availability

The data presented in this study are available on request from the corresponding author. The data are not publicly available because the raw data were generated at the resource center “Physical Methods of Surface Investigation” (RC PMSI) of Research Park of Saint Petersburg State University. Derived data supporting the findings of this study are available from the corresponding author.

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
