# Peer review of "Local Electric Property Modification of Ferroelectric Tunnel Junctions Induced by Variation of Polarization Charge Screening Conditions under Measurement with Scanning Probe Techniques"

_nanomaterials, 2021, doi:10.3390/nano11123323_

Round 1

Reviewer 1 Report

Adreeva, et. al have investigate the local resistive properties of asymmetry FTJs heterstructure with AFM and STM setup and show the importance of polarization charges screening, which largely affect the TER. It's quite interesting and the manuscript is well written. I recommend to publish in Nanomaterials after a minor revision. Some small comments: (1) Line 101, it's fully unstrained on the STO? I think the authors should show the data. (2) Line 240, I am not convinced about the residual water after annealing at 650 degree C, and also in Line 277. After growth of BTO film, the samples were taken out of the chamber for Au depostion? It's bettter to have UHV connnetted chambers for film deposition to aviod the contamination at the interface.

Author Response

Dear Referee,

We highly appreciate your comments and time you spent for reviewing our manuscript. We have taken into account all the critical comments, replied to them, and revised the manuscript accordingly.

Below, we give a detailed, point-by-point reply to all comments. Corresponding changes made in the revised draft of the manuscript are highlighted in red.

Reply to the First Referee

  • Some small comments: (1) Line 101, it's fully unstrained on the STO? I think the authors should show the data.

We agree with the Referee and extend the description of the samples growth in the Materials and Methods section. The following passage is included (pages 3-4, lines 95 - 100):

The deposition process was monitored in situ with Reflection High-Energy Electron Diffraction (RHEED) (see the Supplementary file in Ref. 6). RHEED intensity oscillations indicate a layer-by-layer growth of LSMO. During the growth of BTO films, no RHEED oscillations were observed, but the RHEED pattern remained that of a two-dimensional surface, indicating changes of the growth mode to a step-flow.

In the specified Supplementary file of citation [6] the reader will find detailed information on sample growth and its characterization with RHEED and XRD.

  • Line 240, I am not convinced about the residual water after annealing at 650 degree C, and also in Line 277.

Our statements regarding the remaining water on the sample surface were based on the O 1? XPS spectra taken in Ref. 16 in situ from different oxygen compounds as a function of temperature: “As the top layer is physisorbed OH or H2O, it should be possible to evaporate it, at least partially. Heating a BTO single crystal has a big influence on the physisorbates. A treatment under UHV conditions is necessary so that no new physisorbates form from the moisture in the air. In situ XPS measurements show that above ≈ 350 â—¦C H2O and OH is removed to a large extent (Fig. 7.5). Increasing the temperature up to 800 â—¦C leads to a reduction of the chemisorbed layer” (please, see p. 55. Chapter 7.1 Adsorbates on Ferroelectric Perovskites. 7.1.1. Barium Titanate in Peter, F. Piezoresponse Force Microscopy and Surface Effects of Perovskite Ferroelectric Nanostructures. J. Mater. Sci. 2006, 11. ISSN 1433-5514. ISBN 3-89336-444-7).

However, we definitely made a mistake in the temperature range. T = 650 C is now corrected to T = 350 C (see page 9, line 256). The value of T = 800 C, specified in the text on page 10 (line 297), is related to the chemically adsorbed layer and it is in good agreement with the results of Ref. 16.

  • After growth of BTO film, the samples were taken out of the chamber for Au deposition? It's better to have UHV connected chambers for film deposition to avoid the contamination at the interface.

We completely agree with the Referee. Unfortunately, the top electrodes were patterned through a stencil mask, and we were forced to bring our samples at ambient at this stage. It may be possible to create the entire structure in UHV, but we need to modify the technology by introducing, for instance, the intermediate lithography process or by using some additional manipulators to load the mask into the UHV chamber and place it in a proper position in front of the sample.

With kind regards, on behalf of the authors,

Natalia Andreeva

Reviewer 2 Report

The authors N. Andreeva et al. has reported systematic analysis on the study of TER and IV characteristics from AFM measurements. The manuscript is well written, and it deserves the publication on Nanomaterials. Before that, recommend few clarifications:

  • In line 102 the authors claim that the LSMO and BTO films are fully strained on the STO since the in-plane lattice of the BTO films equals that of the STO substrate. However, no comments on LSMO are mentioned, which it is crucial since it is between both materials.
  • 1c shows brown regions outside the 2 written squares, which are aligned with the inner square. Which is the origin of these regions? Information about the PFM reading conditions must be indicated.
  • In line 109 the authors claim that the screening length for LSMO / BTO interface is equal to 0.1angstroms, according to ref.9. However, in table 1 of that reference it is indicated that the screening length of LSMO is equal to 1angstrom (one order of magnitude higher). This discrepancy must be clarified.
  • In line 219 the authors mention that the small size of the AFM tip results in small current densities. However, I would expect that the current density is the same, but not the current. Some clarification must be provided.
  • In line 243 the authors indicate that there is a drop of 0.195V however I am not able to see this drop. An arrow on the graph may help.
  • In line 255 the authors say that IV curves for the upward polarization lie above the curves for the downward polarized state. However, the current for the upward polarization is lower than for the downward polarization so I suggest to clarify the sentence.
  • There is a typo at line 358 ‘atop’.

Author Response

Dear Referee,

We highly appreciate your comments and time you spent for reviewing our manuscript. We have taken into account all the critical comments, replied to them, and revised the manuscript accordingly.

Below, we give a detailed, point-by-point reply to all comments. Corresponding changes made in the revised draft of the manuscript are highlighted in red.

Reply to the Second Referee

  • In line 102 the authors claim that the LSMO and BTO films are fully strained on the STO since the in-plane lattice of the BTO films equals that of the STO substrate. However, no comments on LSMO are mentioned, which it is crucial since it is between both materials.

We agree with the Referee and extend the description of the samples growth in the Materials and Methods section. The following passage is included (pages 3-4, lines 95 - 100):

The deposition process was monitored in situ with Reflection High-Energy Electron Diffraction (RHEED) (see the Supplementary file in Ref. 6). RHEED intensity oscillations indicate a layer-by-layer growth of LSMO. During the growth of BTO films, no RHEED oscillations were observed, but the RHEED pattern remained that of a two-dimensional surface, indicating changes of the growth mode to a step-flow.

In the specified Supplementary file of citation [6] the reader will find detailed information on sample growth and its characterization with RHEED and XRD.

  • 1c shows brown regions outside the 2 written squares, which are aligned with the inner square. Which is the origin of these regions? Information about the PFM reading conditions must be indicated.

Thank you very much for this comment. We provide the PFM reading conditions in the text of the manuscript (see the caption to the Figure 1). Unfortunately, we put the raw PFM image without filtering the signal properly. While scanning, our AFM setup determines the maximum and the minimum values of the signal at every line of the scan and distributes the range between these values. The resulting image could be confusing. We correct this artifact of PFM phase image representation.

  • In line 109 the authors claim that the screening length for LSMO / BTO interface is equal to 0.1angstroms, according to ref.9. However, in table 1 of that reference it is indicated that the screening length of LSMO is equal to 1angstrom (one order of magnitude higher). This discrepancy must be clarified.

Indeed, in the table 1 of the former reference 9 [Wang, Z.; Zhao, W. A physics-based compact model of ferroelectric tunnel junction for memory and logic design. J. Phys. D: Appl. Phys. 2014, 47, 045001. DOI:10.1088/0022-3727/47/4/045001], the effective screening length for the LSMO / BTO interface is specified as 1 angstrom. However, the imperfect screening in our paper is determined by both effective screening length and dielectric constant according to [Chang, S.-C.; Naeemi, A.; Nikonov, D.E.; Gruverman, A. Theoretical approach to electroresistance in ferroelectric tunnel junctions. Phys. Rev. Applied. 2017, 7, 024005. doi.org/10.1103/PhysRevApplied.7.024005]. The dielectric constant for the LSMO / BTO interface was estimated in that work and could be accepted as 9.8. Thus, in our paper, we consider  . We add the missing reference and the corresponding explanation in the text on pages 4-5 (lines 161-167).

We would like to notice, that in the former reference [9], the effective screening length for the LSMO / BTO interface was taken from the experimental works [Chanthbouala A et al 2011 Solid-state memories based on ferroelectric tunnel junctions Nature Nanotechnol. 7 101 and Chanthbouala A et al 2012 A ferroelectric memristor Nature Mater. 11 860]. In both papers, there are also no direct estimations of the effective screening length at the LSMO / BTO interface, but there is cited the work of [Zhuravlev, M. Y., Sabirianov, R. F., Jaswal, S. S. & Tsymbal, E. Y. Giant electroresistance in ferroelectric tunnel junctions. Phys. Rev. Lett. 94, 246802 (2005)]. In the latter, the screening lengths are estimated in accordance with Thomas-Fermi model.

  • In line 219 the authors mention that the small size of the AFM tip results in small current densities. However, I would expect that the current density is the same, but not the current. Some clarification must be provided.

Thank you very much, we agree with this comment and have corrected the related statement in the manuscript (see page 8, lines 230).

  • In line 243 the authors indicate that there is a drop of 0.195 V however I am not able to see this drop. An arrow on the graph may help.

This voltage drop is schematically shown in Figure 2 (c, d). The voltage drop at the Au top electrode / BTO interface is associated with imperfect polarization in AFM measurements (), and its value was taken as a fitting parameter for the I-V curve modelling. This value was retrieved from the results of approximation with equations 7-9 and could not be pointed with arrows on the graphs.

  • In line 255 the authors say that IV curves for the upward polarization lie above the curves for the downward polarized state. However, the current for the upward polarization is lower than for the downward polarization so I suggest to clarify the sentence.

We agree with the Referee and have corrected this statement accordingly (see page 9, lines 267 - 268).

  • There is a typo at line 358 ‘atop’.

Thank you, the typo is corrected (page 13, line 370).

With kind regards, on behalf of the authors,

Natalia Andreeva
